Large mammal burrows in late Miocene calcic paleosols from central Argentina: paleoenvironment, taphonomy and producers

Cardonatto María Cristina mccardonatto@exactas.unlpam.edu.ar 1
Melchor Ricardo Néstor 2
1 Departamento de Geología, Facultad de Ciencias Exactas y Naturales, Universidad Nacional de La Pampa , Santa Rosa , La Pampa , Argentina
2 Instituto de Ciencias de la Tierra y Ambientales de La Pampa, Universidad Nacional de La Pampa and Consejo Nacional de Investigaciones Científicas y Técnicas , Santa Rosa , La Pampa , Argentina
Marsicano Claudia
Electronic publication date: 2018 May 22
Publication date: 2018
Volume: 6
Electronic Location ID: e4787
Received 2018 Mar 5; Accepted 2018 Apr 27
Copyright: ©2018 Cardonatto and Melchor
Copyright year: 2018
Copyright holder: Cardonatto and Melchor
License: This is an open access article distributed under the terms of the Creative Commons Attribution License, which permits unrestricted use, distribution, reproduction and adaptation in any medium and for any purpose provided that it is properly attributed. For attribution, the original author(s), title, publication source (PeerJ) and either DOI or URL of the article must be cited.
License URL: https://creativecommons.org/licenses/by/4.0/

Keywords: Tetrapod burrow, Taphonomy, Laminated burrow fill, Xenarthra

Funding: PICT 2013-1129 (Agencia Nacional de Promoción Científica y Tecnológica), PIP 2014-2016 11220130100005 (CONICET) Project PI09G from Universidad Nacional de La Pampa to Ricardo Néstor Melchor This work was funded by projects PICT 2013-1129 (Agencia Nacional de Promoción Científica y Tecnológica), PIP 2014-2016 11220130100005 (CONICET), and project PI09G from Universidad Nacional de La Pampa to Ricardo Néstor Melchor. The funders had no role in study design, data collection and analysis, decision to publish, or preparation of the manuscript.

==============================
Large cylindrical sediment-filled structures interpreted as mammal burrows occur within the loess-paleosol sequence of the late Miocene Cerro Azul Formation of central Argentina. A total of 115 burrow fills from three localities were measured. They are typically shallowly dipping, subcylindrical, unbranched structures with rounded ends and lacking enlargements. The horizontal diameter of the burrows range between 0.15 and 1.50 m, with most of the burrows in the interval of 0.39 to 0.98 m. Geometric morphometric analysis of transverse cross-sections support their distinct subcircular and elliptical (horizontally flattened) shapes. Burrow fills are typically laminated in the lower part and massive in the upper part. The laminated intervals reflect pulses of flowing water entering the abandoned burrow during moderate rains, whereas massive intervals reflect mass flow input of dense sediment-water mixtures during heavy rains that produced sheet floods. Approximately 1% of the burrows contained fragmentary, disarticulated and weathered mammal bones that were introduced in the open burrow by currents along with other sedimentary particles. Analysis of the tetrapod burrow fossil record suggests that Miocene burrows, including those studied herein, reflect a remarkable increase in the average size of the fossorial fauna. We conclude that large late Miocene mammals dug burrows essentially as a shelter against environmental extremes and to escape predation. The simple architecture of the burrows suggests that the producers essentially foraged aboveground. Several mammal groups acquired fossorial habits in response to cold and seasonally dry climatic conditions that prevailed during the late Miocene in southern South America. The considerable range of horizontal diameters of the studied burrows can be attributed to a variety of producers, including dasypodids, the notoungulate Paedotherium minor, Glyptodontidae and Proscelidodon sp.

Introduction

Fossil vertebrate burrows are relatively common biogenic structures and the oldest convincing evidence of tetrapod burrows are Early Permian (Asselian-Artinskian) lysophorid amphibian burrows from Kansas, USA (Hembree, Martin & Hasiotis, 2004). Most pre-Cretaceous tetrapod burrows have been attributed to therapsids, in part on the basis of the finding of articulated skeletons in a few Late Permian-Early Triassic burrows (Smith, 1987; Groenewald, Welman & MacEachern, 2001; Damiani et al., 2003; Modesto & Botha Brink, 2010). A common architecture for Permian to Jurassic tetrapod burrows is a shallowly inclined ramp with a rounded and not enlarged end, of reduced horizontal diameter (Fig. 1), with discrete scratch marks, always showing a horizontally flattened elliptical cross-section, and commonly with a bilobed bottom (e.g., Damiani et al., 2003; Sidor, Miller & Isbell, 2008; Riese, Hasiotis & Odier, 2011; Liu & Li, 2013; Melchor & Loope, 2016; Krummeck & Bordy, 2018). Most of Jurassic tetrapod burrows occur in eolian sequences including the oldest subhorizontal burrow systems that have been assigned to primitive mammals from the Early Jurassic Navajo Sandstone of USA (Riese, Hasiotis & Odier, 2011) (Fig. 1). In contrast, there is a dearth of reports of Cretaceous tetrapod burrows which could possibly be due to the more equable climates that existed for most of this period. An exceptional occurrence for the Late Cretaceous is the unique dinosaur burrow containing an adult and two juvenile remains of ornithopods, further suggesting denning behavior and parental care (Varricchio, Martin & Katsura, 2007).

Figure 1 Plot of horizontal diameter of fossil tetrapod burrows vs age.

Plot of the average horizontal diameter (Dh) of fossil tetrapod burrows in the published literature against the age of the hosting rock, distinguishing between burrows with remains that were interpreted as belonging to the producer, those lacking bone remains and the present study. Note that the age axis contains two gaps in the Cretaceous and Paleocene. Source of information on Article S1. Image credit: Ricardo N. Melchor and María C. Cardonatto.

Most Cenozoic tetrapod burrows have been attributed to mammals, mainly to Rodentia and Xenarthra (e.g., Voorhies, 1975; Martin & Bennett, 1977; Benton, 1988; Gobetz & Martin, 2006), whereas a few examples are related to Carnivora (e.g., Hunt, Xiang-Xu & Kaufman, 1983; Hembree & Hasiotis, 2008). The record of Paleogene tetrapod burrows is meager and may also be linked to dominantly benign climate conditions (Fig. 1). The Miocene record of tetrapod burrows is more varied and abundant, with a diversification of the architectural patterns and behavioral strategies that, commonly, appeared under stressed volcaniclastic and eolian environments. The early Miocene volcaniclastic floodplains of Nebraska, USA, witnessed the appearance of (1) the celebrated vertical helical burrows ending in a ramp and terminal chamber named Daimonelix (e.g., Barbour, 1892; Martin & Bennett, 1977), as well as (2) smaller, complex subhorizontal rodent burrow systems with terminal chambers and subcircular cross-section (Gobetz & Martin, 2006), and (3) the first carnivore den (Hunt, Xiang-Xu & Kaufman, 1983; Hunt, 1990). Also in the early Miocene, the coastal dunes of Germany preserved the oldest fossil food cache (Gee, Sander & Petzelberger, 2003). The main tetrapod burrowing innovation during the middle Miocene is represented by cylindrical, subhorizontal, unbranched tunnels with a meniscate backfill interpreted as foraging tunnels of small Dasypodidae from southern South America (Melchor et al., 2012; Melchor et al., 2016). Pliocene burrows are small (horizontal diameter less than 250 mm) and those from the Atlantic coast of the Buenos Aires province, Argentina have received a considerable attention, because they are common and a significant number of the burrows contains bone remains (e.g., Genise, 1989; Quintana, 1992; Fernández, Vassallo & Zárate, 2000; Elissamburu, Dondas & De Santis, 2011). These are subcircular burrows assigned to rodents and notoungulates. The Pleistocene megafauna of South America is also reflected in the burrow trace fossil record in the form of huge tunnels (up to 2 m wide), with horizontally flattened elliptical cross-sections from Argentina and Brazil (e.g., Quintana, 1992; Vizcaíno et al., 2001, Buchmann, Pereira Lopez & Caron, 2009, Genise & Farina, 2012; Frank et al., 2012; Frank et al., 2015). The smaller burrows are attributed to Dasypodidae and Pampatheriidae and the larger to ground sloths. It has been suggested that the adoption or generalization of burrowing behavior by large Pleistocene herbivorous mammals may reflect the arrival of large carnivorans after the Great American Biotic Interchange, just before the Pliocene-Pleistocene boundary (Soibelzon et al., 2009).

A trend towards larger diameter burrows is evident from the early Miocene to the late Pleistocene (see Fig. 1). Pre-Miocene burrows attain a maximum average horizontal diameter (Dh) of 420 mm (the examples from the Navajo Sandstone described by Loope, 2006); however, most are in the range of 100–200 mm. From the Miocene on, burrows with an average horizontal diameter in excess of 500 mm are recorded (Fig. 1), including those studied herein. The Neogene record also includes more common and smaller burrows (Dh ∼  < 200 mm) that are mostly attributed to rodents and small Dasypodidae (Fig. 1).

In this context, the tetrapod burrows from the late Miocene Cerro Azul Formation are the largest pre-Pleistocene tetrapod burrows and can help to understand the reasons for acquisition of burrowing habits in large Cenozoic mammals from South America. These structures have been partially and briefly described (Genise et al., 2013), but a detailed description has yet to be done and is an objective of this study. The purposes of this work are: (1) to infer the likely producers of these late Miocene large burrows and (2) to interpret the taphonomic processes involved in the preservation of the burrow casts and its paleoecological and paleoenvironmental meaning.

Material & Methods

The studied localities with late Miocene burrows are distributed in a latitudinal belt of approximately 25 km (Fig. 2): Salinas Grandes de Hidalgo (SG, 37°12′55″S, 63°35′25″W; 100 masl); Laguna Chillhué (LC, 37°19′15.13″S; 64°14′31.52″W; 145 masl); and Laguna La Paraguaya near Carhué city (LLP, 37°5′53.57″S; 62°47′34.98″W; 101 masl). The first two localities belong to La Pampa province, whereas the remaining is located in the adjacent Buenos Aires province. The burrow fills appear in outcrop as transverse to oblique, longitudinal, tangential and plan view exposures.

Figure 2 Study area.

Study area in central Argentina (inset), outcrops of the Cerro Azul Formation and localities in La Pampa and Buenos Aires provinces. LC: Laguna Chillhué, SG: Salinas Grandes de Hidalgo, LLP: Laguna La Paraguaya. Modified from Cardonatto et al. (2016).

Another locality of the Cerro Azul Formation, a roadcut in national road 154 (R154, 37°49′28.5″S, 64°4′8.9″W), has been previously described as having vertebrate burrows (De Elorriaga & Visconti, 2001). However, this locality is not considered herein because the burrows probably postdate significantly the deposition of the unit. The reasons for this inference are: (1) burrow diameters are considerably larger than those described herein (Dv up to 2.25 m) and more consistent with burrows attributed to a large Pleistocene megafauna (e.g., Vizcaíno et al., 2001; Frank et al., 2012); and (2) the burrow cut the carbonate nodules of the capping calcrete. Although absolute dating of the calcrete is not available, it has been suggested that the calcretization process significantly postdates the deposition of the Cerro Azul Formation (Vogt, Carballo & Calmels, 1999; Melchor & Casadío, 2000; Visconti et al., 2010). The main argument for this inference is that the calcrete is developed in sedimentary sequences ranging in age from late Miocene to Pleistocene.

Standard measurements in burrows were horizontal diameter (Dh), vertical diameter (Dv), preserved length, mean azimuth of burrow fill and inclination of fill laminae. The mean azimuth was measured using a compass and considering the burrow fill boundary and dominant plunge of laminated fill. When changes of dip direction or inclination were observed in a single burrow fill they were recorded separately. Burrow diameter (especially Dh) was measured orthogonal to the main axis of the structure. The horizontal diameter was obtained from almost all types of exposures (except for longitudinal ones), whereas the vertical diameter was mostly an apparent value, except for the rare transverse sections, where it can be considered the true vertical diameter. At each location, GPS coordinates were recorded; the burrow fills were photographed and sketched.

The burrow fills exposed in transverse section (n = 24) were used for a 2-D geometric morphometric analysis. From field photographs, the outline of the burrow boundary was sketched in Corel Draw™. These images were used to build a file with the TPSutil software. Burrow cross section outlines were oriented with respect to the top and bottom of the hosting bed and treated as symmetrical outlines. For each image a total of 16 type II landmarks (Slice et al., 2008) were digitised using the TPSdig2 software- Landmark 1 and 9 were positioned in the intersection of the burrow outline and the maximum vertical axis, whereas landmarks 5 and 13 resulted from the intersection of the maximum horizontal axis with the burrow outline. The remaining landmars were arranged with an equidistant pattern on the burrow outline (Fig. 3 and Article S2). Landmarks were aligned using the Procrustes superposition method (MacLeod, 2009) and the principal component analyses (PCA) using MorphoJ software. Results are presented by eigenvalue diagrams and PC scores, along with transformation grids.

Figure 3 Example of location of landmarks.

Location of landmarks (red points) on burrow fills preserved in cross section. Photo credit: Ricardo Néstor Melchor.

From scaled sketches of transverse sections of burrow fills, the cross-sectional area of the burrow was estimated using ImageJ software. This cross-sectional area was used to estimate the body mass of the producer using the allometric relations of Wu et al. (2015).

At each locality, a detailed sedimentary log was measured and samples of host rock and burrow fill were collected for petrographic analysis. Also, a selected locations a detailed sedimentologic log of the burrow fill was measured. Mammal bones found in situ within burrow fills were recorded and collected for preparation and taxonomic identification. Fossil bones found outside burrows were not recorded or collected. Fossil remains from La Pampa province collected during this study are housed at the Paleontological Collection of the Facultad de Ciencias Exactas y Naturales, Universidad Nacional de La Pampa, Santa Rosa city, La Pampa, Argentina, under the acronym GHUNLPam. Fossil material collected in burrows from Laguna La Paraguaya locality (Buenos Aires province) are housed at the Museo Histórico Regional de Guaminí “Coronel Marcelino E. Freyre” under the acronym MHG-P. Field work was approved by the Dirección Provincial de Museos y Preservación Patrimonial, under the Project “Vertebrados del Mioceno tardío-Plioceno en el área de las lagunas encadenadas del oeste de la provincia de Buenos Aires. Aportes a la bioestratigrafía del Cenozoico tardío de la Región Pampeana”, permit nr. 2015-3-P-156-2.

Geological setting

The Cerro Azul Formation outcrops are located in the north-western, central, and eastern part of La Pampa province and adjacent western Buenos Aires province, Argentina (Linares, Llambías & Latorre, 1980; Folguera & Zárate, 2009; Visconti et al., 2010). The unit is characterized by a monotonous succession of loess containing moderately developed paleosols (Fig. 4) that has been assigned to the late Miocene (Huayquerian Land Mammal age) essentially on the basis of its mammal remains (Montalvo & Casadío, 1988; Verzi, Montalvo & Vucetich, 1999; Verzi, Montalvo & Tiranti, 2003). In particular, the formation is considered as representing the interval between 10 and 5.7 Ma (Cione et al., 2000; Verzi, Montalvo & Deschamps, 2008). The maximum exposed thickness in outcrop is 54 m, although the unit reaches about 180 m in the subsurface (Visconti et al., 2010). The formation is essentially composed of structureless, light brown (5YR 6/4), pale reddish brown (10R 5/4) or grayish orange pink (5YR 7/2), sandy siltstones and fine-grained sandstones, showing moderate selection and common carbonate cementation.

Figure 4 Sedimentary logs.

Detailed sedimentary logs of the measured sections from the study localities. Image credit: Ricardo Néstor Melchor.

Results

Sedimentology of the burrow—bearing sections

Macroscopic features of the studied sections are very similar to those of the classical localities of the Cerro Azul Formation outcrops (e.g., Visconti et al., 2010; Genise et al., 2013; Cardonatto et al., 2016), especially those of the Salinas Grandes de Hidalgo and Laguna La Paraguaya. Paleosol profiles are typically composed of two horizons (Fig. 4). The upper horizon is a clayey siltstone that is distinguished by the presence of subangular blocky, granular or prismatic peds and a darker color (5 YR 6/4) than the underlying horizon. Carbonates are rare except for the local occurrence of calcareous rhizoliths. This upper horizon can be compared with a Bt horizon and its thickness averages 0.60 m (range = 0.35–1.00 m, n = 5), which is very close to the average for the formation (Cardonatto et al., 2016).

The lower horizon is characterized by lighter colored (5 YR 7/2), siltstone to fine- grained sandstone with pervasive carbonate cementation, both in the form of nodules and rhizoliths. Rhizoliths are small to medium sized and commonly 1–30 mm in diameter. The lower horizon can be up to more than 5 m thick and contain relicts of primary sedimentary structures, like tangential cross-bedding or horizontal bedding, as seen in the Salinas Grandes de Hidalgo section (Fig. 4). Mudstone intraclasts are common throughout. The remains of primary sedimentary structures and carbonate cementation suggest comparison with a Bk or Bk/C horizon. The trace fossils described in this paper occur in both horizons.

The section at Laguna Chillhué also contains similar paleosols (Fig. 4) and differs from the other localities by the presence of a 2 m thick, dark-red, laminated mudstone interval in the lower part of the section (Montalvo et al., 1995). The presence of a mudstone interval in the lower section of the Cerro Azul Formation has been questioned by Lorenzo, Mehl & Zárate (2013), who supposed a younger age for this mudstone interval on the basis of geomorphological inferences. However, at this location the laminated mudstone interval is overlain, through a normal sedimentary contact, by lithologies typical of the Cerro Azul Formation containing mammal remaims of Huayquerian (late Miocene) age, with no evidence of reworking. Vertebrate burrow fills were not observed in the laminated mudstone interval.

Description of large burrow fills

A total of 115 fossil burrows were measured from three localities: 53 from Salinas Grandes de Hidalgo, 59 from Laguna La Paraguaya, and three from Laguna Chillhué (see Table S1). The studied burrow fills are distinguished on the basis of the presence of a thinly laminated siltstone to mudstone interval that contrasts with the structureless host rock. When the upper part of the fill is massive and similar in grain size to the host rock, the upper burrow boundary is indistinct. Burrow fills exhibit an induration, composition and cementation similar to the hosting rocks, suggesting that they are too of late Miocene age. At Laguna La Paraguaya locality, the preferential carbonate cementation of the burrow fills resulted in 3-D exposures (Fig. 5A). At this locality the density of fossil burrows is locally high and may be difficult to find unburrowed intervals. Cross-cutting relationships between two or more burrows are common (Figs. 5B, 5C).

Figure 5 Abundance and cross-cutting relationships of burrows, from LLP locality.

(A) General view of the outcrop showing several burrows (yellow arrows). (B–C) Field view and diagram of cross-cutting relationships between different fossil burrows (distinguished in C with different shades of gray). Photo and image credit: Ricardo Néstor Melchor.

Size and plan view pattern

Observed horizontal diameter (Dh) ranges from 0.15 to 1.50 m ( n = 112) and the frequency distribution suggests a roughly normal distribution where three subpopulations can be distinguished (Fig. 6). The small subpopulation has a Dh from 0.15 to 0.34 m (8 %), the intermediate subpopulation has a Dh from 0.39 to 0.98 m (84 %), and the large subpopulation exhibits a Dh from 1.05 to 1.50 m (8 %).

Figure 6 Histogram of horizontal diameter.

Histogram showing the frequency distribution of horizontal diameter (Dh) for the studied fossil burrows. Three subpopulations can be distinguished. Image credit: María Cristina Cardonatto.

In plan view exposure, which is found only at SG and LLP localities (n = 78), a number of morphologies can be distinguished (Fig. 7). (1) The more common are straight to slightly curved burrows (89 % of cases), which exhibits a Dh= 0.15–1.15 m, showing a uniform inclination of internal laminae (ranging from ≈ 0° to 27°), the maximum height difference between the proximal and distal portion of a burrow is 0.6 m, and the maximum preserved length is 5.18 m (Figs. 7A, 7B). Some burrow fills in this category display a decrease in inclination of internal laminae toward more distal positions (i.e., from 27° to 8°). (2) A sinuous burrow that displays two opposite curves in plan view was recorded in 5% of the cases (Figs. 7C, 7D). The horizontal diameter of sinuous burrow ranges from 0.42 to 0.80 m, dip of internal laminae is subhorizontal to slightly inclined (up to 8°), and the maximum observed length is 8 m. (3) The third plan view pattern is a C-shaped curve observed in 6% of the burrows, with an horizontal diameter ranging from 0.44 to 0.72 m (Figs. 7E 7F), which commonly appears as a ramp with a height difference of up to 0.55 m, the inclination of internal laminae can be uniform (from 3°to 12°) or show a shallowing toward the distal position (from 14° to subhorizontal).

Figure 7 Burrow morphology in plan view.

(A–B) Field photograph and diagram of slightly curved burrow. (C–D) Field photograph and diagram of sinuous burrow. (E–F) Field photograph and diagram of “C” shaped burrow. (G–H) Field photograph and diagram of burrow with slightly enlarged and rounded end. Curved lines in the diagrams represent weathered laminae. Photo credit: Ricardo Néstor Melchor. Image credit: María Cristina Cardonatto.

In a few cases, the distal portion of burrow showed a lateral expansion of up to 23% of the Dh, commonly having a subhorizontal lamination (Figs. 7G, 7H). Other burrow fills exhibit a rounded end with no enlargement that can be accompanied by an upward bending of mudstone laminae against the walls of the burrow.

Cross-sectional shape and body mass

The analysis of the well defined cross-sectional shape of burrows (n = 24 from all localities) suggest a distinction between elliptical (with the major axis subhorizontal) and subcircular cross-sections. Elliptical cross-sections are more common (n = 18) and the corresponding Dh ranges from 0.39 to 1.50 m (belonging to the intermediate and large subpopulations, Fig. 6), with an average Dv/Dh ratio of 0.55. The burrows with elliptical cross-section include a few cases (n = 4) with a flat bottom and convex top. The subcircular cross-sectional shape (n = 6) is represented in the intermediate subpopulation with a Dh ranging from 0.39 to 0.56 m, and an average Dv/Dh ratio of 0.88.

Morphometric analyses suggest that 90.13 % of the variation is explained by the first two principal components (Fig. 8B), and deformation grids range from elliptical (score = −0.12) to subcircular (score = 0.17) (Fig. 8A).

Figure 8 Results of geometric morphometric analysis of fossil burrows preserved in cross section.

(A) Plot of principal components 1 and 2, distinguishing by study locality, and deformation grids for elliptical (PC1 score −0.12) and subcircular (PC1 score 0.17) shapes. (B) Histogram of variance of principal components. Image credit: María Cristina Cardonatto.

Body mass estimates of the producers of the burrow on the basis of the cross-sectional area (using the method by Wu et al., 2015) suggest that there are two ranges (Table S1). Most of the estimates (n = 18) belongs to the intermediate subpopulation with a range from 37 to 439 kg, whereas the remaining estimates comes from the large subpopulation (n = 7) with a range 708 to 1,623 kg. Burrows with subcircular cross-section from the intermediate subpopulation, are linked with a producer having body mass from 92.84 to 186.0 kg.

Orientation and inclination

Readings of plunge azimuths of burrow fills from all localities are variable but most values are located in the northeast to southeast quadrants (i.e., between N20° and N140°) (Figs. 9A, 9C). The average dip angle of all measured burrows with respect to the paleohorizontal is 7.25°and ranges from nearly 0 to 27° (Fig. 9B). Most orientation data come from the intermediate subpopulation (Dh = 0.39–0.98 m) and especially from LLP locality.

Figure 9 Orientation of fossil burrows compared with modern data from dasypodid burrows.

(A) Rose diagram showing the dip azimuth of fossil burrows. (B) Equal area projection of dip azimuth and dip angle of fossil burrows. (C) Entrance orientation of fossil burrows, assuming that it is located at 180°of measured dip azimuth. The data in A–C is from SG and LLP localities, those from the remaining locality are preserved only in cross-section. (D) Entrance orientation of several species of dasypodid burrows from semiarid settings of northern-central Argentina build from data by Crespo (1944). (E) Entrance orientation of Dasypus hibridus burrows from a grassland setting in Uruguay build from data by González, Soutullo & Altuna (2001). (F) Entrance orientation of Chaetophractus villosus burrows from cultivated land in Buenos Aires province, Argentina build from data by Abba, Udrizar & Vizcaíno (2005). Image credit: Ricardo Néstor Melchor.

Composition of burrow fills

The vertebrate burrows are easily spotted in the field because of the distinctive laminated structure of the infilling sediments that contrasts with the structureless host rock. The filling is composed of an alternation of laminated claystone and siltstone with massive fine-grained sandstone and siltstone containing floating claystone intraclasts. Laminated intervals are a few millimeters to about 50 mm thick, whereas massive intervals tend to be thicker. Most of the burrow fills display a laminated interval in the lowermost part of the fill, with the upper part massive, especially in the Salinas Grandes de Hidalgo locality (Figs. 10A–10C). A few burrows display a poorly defined lamination to massive structure throughout (Fig. 10D). Claystone and siltstone laminae at the bottom of the structure typically deflect upwards against the burrow wall, which is a good criterion to distinguish burrow fills that are mostly exhumed by erosion. Laminae tend to be horizontal but successive laminated packages resting at low angles were also identified. Individual laminae are normally graded (typically siltstone grading to claystone), and locally disrupted giving a brecciated aspect. Both synsedimentary faulting and deformation were identified (Fig. 10F). A pseudomeniscate structure was identified in two cases (one from Salinas Grandes de Hidalgo and the other from Laguna La Paraguaya). This structure is composed of massive siltstone or fine-grained sandstone arranged in adjacent crescent-shaped bodies with the convex margin pointing downslope that span the full width of the fill (Fig. 10E). Horizontal width of individual pseudomeniscate bodies taken parallel to the burrow axis is 120 mm.

Figure 10 Features of fossil burrow fills.

(A–B) Detailed sedimentary logs of the fill of selected burrows (see Table S1 for information on these burrow fills). References as for Fig. 4. (C) Cross-section of laminated to massive burrow fill # 648 from SG, represented in (B). Elliptical outline of fill indicated by yellow arrows, black arrow point to subvertical rhizolith cutting the laminated fill. (D) Cross-section of burrow fill # 714 from LLP. Subcircular outline of poorly laminated to massive burrow fill indicated by yellow arrows. (E) Pseudomeniscate structure in burrow fill # 704 from LLP seen in plan view. Yellow arrows point the outline of the burrow and black arrows to individual pseudomenisci. (F) Detail of laminated burrow fill (# 632 from SG) showing direct grading in siltstone to claystone laminae (yellow arrows), synsedimentary fault (white arrows) and onlap of clay laminae on burrow wall (black arrow). Image credit: Ricardo Néstor Melchor.

Associated ichnofossils

Only the ichnofossils found within or very close to the burrow fills are considered. We found within the fills abundant rhizoliths and rare smaller vertebrate burrows, vertebrate footprints and ?Rosellichnus isp. In the paleosol adjacent to the fills we found Taenidium barretti, Attaichnus kuenzelli and Coprinisphaera isp.

Calcareous rhizoliths, including rhizocretions and root casts, are abundant and were recorded in all studied localities. Rhizoliths are arranged in vertical, oblique and subhorizontal position (Fig. 10C). Rhizoliths are submillimeter to 30 mm thick and the maximum preserved length is 500 mm. Rhizoliths occurring inside burrow fills are similar in shape and cementation to those of the hosting rock.

A single small burrow (Fig. 11A) that cuts the laminated interval of a larger vertebrate burrow fill was identified at Salinas Grandes de Hidalgo (# 638). The 43 mm in diameter structure displays a subcircular outline and a poorly laminated siltstone fill.

Figure 11 Ichnofossils associated with burrow fills.

(A) Small burrow fill (black arrows) cutting the laminated fill of a larger mammal burrow from SG locality. (B) Coprinisphaera isp. from LLP locality. (C) Tetrapod footprints on the top of laminated fill of a burrow from LLP locality. Numbers refers to individual footprints. (D) ?Rosellichnus isp. (arrowed) inside a burrow fill from SG. Photo credit: Ricardo Néstor Melchor.

A partially eroded burrow fill from Laguna La Paraguaya (# 708) exposed an internal bedding plane of the filling showing closely spaced elliptical depressions with a noticeable marginal rim (Fig. 11C). These are tentatively interpreted as footprints of a quadrupedal animal composing a 316 mm wide trackway. If this is a trackway a pace angulation of 112° and a stride length of 600 mm can be inferred. Measurements on individual footprints indicate that average footprint length is 144 mm, average footprint width is 93 mm, and the marginal rim is of uniform thickness and about 50 mm wide.

A group of five subcircular rings in the upper part of a burrow fill (# 659A from SG) is tentatively identified as a cluster of bee cells and compared with the ichnogenus Rosellichnus (Fig. 11D). The presence of the ichnogenus at this locality, although at other section, was already documented by Cardonatto et al. (2016).

Adjacent to the burrow fills at Salinas Grandes de Hidalgo, several specimens of Attaichnus kuenzelii were identified, in some cases very close, but no cross-cutting relationship were seen. A few specimens of Taenidium barretti also occur at this locality, in the form of subcylindrical burrows, 12 mm wide and 80 mm long with an average meniscus thickness of 2.2 mm. At Laguna La Paraguaya we also found two specimens of cemented and compact spherical chambers (diameter 18.7–23.0 mm) with a large emergence hole (10–11 mm) assigned to Coprinisphaera isp. (Fig. 11B).

Bone remains found within burrow fills

Mammal bones within the burrow fills are scarce (only 1% of burrows contained fossil remains) and usually appear disarticulated and poorly preserved, in some cases with signs of abrasion (Fig. 12A). The fossil remains from the SG locality are Proscelidodon sp. and Glyptodontidae indet., whereas the rest of the fossil material was found at LLP locality, including: Paedotherium minor (two specimens), Doellotatus sp., Eosclerocalyptus sp., Mesotheriinae indet., Gyptodontidae indet. (three specimens), and undeterminate mammals (two specimens). For details about the taxonomy and illustrations of mammal remains, see Article S3 and Fig. S1. The only articulated remains are glyptodont osteoderms found at Laguna La Paraguaya (# 702) that are assigned to Eosclerocalyptus sp. (Fig. 12B), and remains of a carapace with several articulated osteoderms from the same locality (# 670) assigned to Glyptodontidae indet. Proscelidodon sp. remains (including a hemimandible with teeth and postcranial elements) appeared disarticulated but associated within a single burrow fill. The fossil remains display different degree of weathering and corrosion, as well as biostratinomic fractures.

Figure 12 Bone remains inside burrow fills.

(A) Close-up of isolated, weathered and fragmentary glyptodontid osteoderms from a vertical section of a burrow fill from SG locality. The burrow boundary is not shown in the photograph. (B) Partly articulated osteoderms of Eosclerocalyptus sp. found inside a burrow fill from LLP locality. Photo credit: Ricardo Néstor Melchor.

Discussion

Producers

The studied fossil burrows are unbranched and display a significant variation in the horizontal diameter, which ranges from 0.15 m to 1.5 m (Fig. 6). The simple, ramp type morphology of the studied burrows suggests that the animals foraged aboveground (e.g., Reichman & Smith, 1990). In order to infer the likely producers of the fossil burrows there are several constraints that need to be considered: (1) the faunal remains found inside the burrow fills; (2) the fossorial mammals that were recorded in the Cerro Azul Formation, especially those from the studied localities; (3) the size of burrows, as expressed by the Dh; and (4) the overall architecture and cross-section of burrows (including the Dv/Dh ratio) and the extrapolated body mass of its digger. The surface ornamentation of burrows is commonly a very useful clue to the producer (e.g., Seilacher, 2007); however, it is not preserved in the studied cases.

Faunal remains found in burrow fillings

In general, bone remains found inside a burrow can be considered as belonging to its producer or occupant only if they are articulated, disarticulated but still closely associated, nearly complete, are commonly found in a terminal portion and fit the size (cross-sectional diameter) of the burrow (e.g., Smith, 1987; Groenewald, Welman & MacEachern, 2001; Damiani et al., 2003). The remains found inside the studied burrows do not fulfill any of these criteria. In most cases, these bone remains have been passively introduced and it is uncertain if they belong to the producers. The remains are essentially fragmentary, disarticulated, with evidence for abrasion and weathering (Fig. 12A, Fig. S1); suggesting that they spent some time at the surface and then were introduced into the burrows by currents along with other sedimentary particles. The fragmentary and disarticulated state of Doellotatus sp. and one of the specimens of Paedotherium minor and the considerably small size of the animals (body mass about 1–2 kg, Table 1) in comparison with the containing burrows; further suggest that these remains were introduced by currents. In the case of Proscelidodon sp., the bones are disarticulated but associated, which suggest that they can belong to a single specimen, and the partial horizontal diameter of the burrow match the size of this ground sloth. The only articulated remains are fragments of the dorsal carapace of Glyptodontidae that occur in burrows large enough to hug these animals (Dh = 0.78 to 1.50 m) (Table 1). In consequence, the unique remains that can belong to the producer of the burrows are Proscelidodon sp. and those of Glyptodontidae.

Table 1 Body mass estimate of producers.

Relationship between cross-sections and body mass of the putative producers, estimated body mass according to Wu et al. (2015). See estimate of cross-sectional area and body mass for every burrow in Table S1.

Range of burrow Dh (m)	Estimaded body mass (kg)	Fossil remains inside burrow fill	Potential burrow producer	Body mass of potential producer (kg)	
0.15–0.34	1–13	Paedotherium minor	Paedotherium	1.86 (Elissamburu, 2004)	
			Doellotatus	Less than 1 (Vizcaíno & Fariña, 1999)	
			Chasicotatus	Less than 1 (Scillato-Yané, Krmpotic & Esteban, 2010)	
			Proeuphractus	2–3 (Perea & Scillato-Yané, 1995)	
			Chorobates	1–10 (Vizcaíno & Fariña, 1999)	
			Lagostomus	1–10 (Vizcaíno & Fariña, 1999)	
0.39–0.94	37–439	Mesotheriinae indet.	Mesotheriinae	20.88–60.13 (Croft, Flynn & Wyss, 2004)	
		Eosclerocalyptus sp.	Eosclerocalyptus	More than 100 (Vizcaíno & Fariña, 1999)	
		Gliptodontidae indet.	Coscinocercus	More than 100 (Vizcaíno & Fariña, 1999)	
		Gliptodontidae indet.	Aspidocalyptus	More than 100 (Vizcaíno & Fariña, 1999)	
			Macrochorobates	10–100 (Vizcaíno & Fariña, 1999)	
			Macroeuphractus	10–100 (Vizcaíno & Fariña, 1999)	
		Proscelidodon sp.	Proscelidodon	581.8 (De Esteban-Trivigno, Mendoza & De Renzi, 2008); 850 (Bargo et al., 2000); 1,057 (Fariña, Vizcaíno & Bargo, 1998). Body mass of S. leptocephalum	
		Paedotherium minor			
		Doellotatus sp.			
1.05–1.5	708–1,623	Gliptodontidae indet.	Glyptodontidae	More than 100 (Vizcaíno & Fariña, 1999)	
			Proscelidodon	581.8 (De Esteban-Trivigno, Mendoza & De Renzi, 2008); 850 (Bargo et al., 2000); 1,057 (Fariña, Vizcaíno & Bargo, 1998). Body mass of S. leptocephalum.	
Notes.

Dh, horizontal diameter.

Fossorial mammals of the Cerro Azul Formation and size of burrows

The mammals with fossorial habits recorded in the Cerro Azul Formation include xenarthrans, notoungulates and rodents (e.g., Goin, Montalvo & Visconti, 2000; Cerdeño & Montalvo, 2001; Urrutia, Montalvo & Scilato-Yané, 2008). Among the Xenarthra, the Glyptodontidae, Dasypodidae and Mylodontidae display fossorial adaptations. The same is true for Mesotheridae and Hegetotheriidae (Notoungulata); and Caviidae, Octodontidae, and Chinchillidae (Rodentia). Below we discuss the potential producers for each size class of the burrows (Table 1) as expressed by the horizontal diameter and cross-sectional area of the burrows.

For the small subpopulation (Dh = 0.15–0.34 m, 8% of cases), with a body mass ranging from 1 to 13 kg, the likely candidates are the notoungulate Paedotherium minor, the dasypodids Doellotatus, Chorobates, Proeuphractus, and Chasicotatus; and the rodent Lagostomus. Paedotherium (Hegetotheriidae) is a medium-sized rodent-like ungulate native to South America. This taxon is very common in the Cerro Azul Formation, both in La Pampa and Buenos Aires provinces (e.g., Montalvo, Tomassini & Sostillo, 2016). Articulated remains of this genus have been found within Pliocene burrow casts (about 0.16–0.22 m wide) from the Atlantic coast of Buenos Aires province (e.g., Genise, 1989; Scognamillo, 1993; Elissamburu, Dondas & De Santis, 2011) and a morphofunctional analysis of its postcranial skeleton suggest a digging capacity (Elissamburu, 2004).

The Dasypodidae show a neotropical geographic distribution and were important components of the late Miocene-Pliocene South American fauna (Scillato-Yané, 1982; Ortiz Jaureguizar, 1998). Dasypodids exhibit fossorial habits and were abundant during the late Miocene in the Pampean region of Argentina, suggesting preference for open environments and well drained soils (Scillato-Yané et al., 2013). Most dasypodids recorded in the Cerro Azul Formation were small- to medium-sized, with body mass in the range 1–10 kg for Doellotatus, Chasicotatus, Proeuphractus and Chorobates (Table 1). In particular, the holotype of Chasicotatus ameghinoi is a nearly complete carapace about 150 mm wide (Scillato-Yané, Krmpotic & Esteban, 2010), which match the lower size range of the small subpopulation. Modern dasypodid burrows are usually simple ramps lacking significant enlargements (e.g., González, Soutullo & Altuna, 2001; Abba, Udrizar & Vizcaíno, 2005), which is similar to the architecture of the fossil burrows.

In the same localities of Paedotherium-bearing burrows from the Atlantic coast of the Buenos Aires province, there are also burrows containing articulated remains of Lagostomus that partially overlap in diameter with those containing Paedotherium remains (Genise, 1989; Elissamburu, Dondas & De Santis, 2011). The extant Lagostomus maximus (plains vizcacha) is well known for its digging adaptations and for living in communal burrow systems (e.g., Jackson, Branch & Villarreal, 1996). Plains vizcacha burrow systems show an average entrance horizontal diameter of 0.26 m and a range of 0.17–0.37 m (Llanos & Crespo, 1952), which matches the range of the small subpopulation. However, extant L. maximus burrow systems have several entrance ramps that typically converge into a central chamber or a much more complex architecture (e.g., Llanos & Crespo, 1952; Rafuse et al., in press), which contrast with the simple ramp type morphology of the fossil burrows. The 43 mm in diameter subcircular burrow identified in the fill of a larger burrow at Salinas Grandes de Hidalgo (# 638) is probably related to a caviomorph rodent (Caviidae or Octodontidae).

For the dominant intermediate subpopulation (Dh = 0.39–0.94 m, 83% of measured burrows), with an estimated body mass ranging from 37 to 438 kg, the likely candidates are the Mesotheriinae (Mesotheriidae, Notoungulata); Eosclerocalyptus, Coscinocercus, and Aspidocalyptus (Xenarthra, Glyptodontidae); Macrochorobates and Macroeuphractus (Xenarthra, Dasypodidae); and Proscelidodon (Xenarthra, Mylodontidae). The fossil remains found in this size range that are likely candidates are those of Glyptodontidae and Proscelidodon sp. (Table 1). There are two Mesotheriinae species recognized for the late Miocene of central Argentina: Pseudotypotherium subinsigne and Typotheriopsis silveyrai (Cerdeño & Montalvo, 2001). These species exhibited a small to medium size (20.88 to 60.13 kg after Croft, Flynn & Wyss, 2004) (Table 1). The Mesotheriidae shows modifications in the appendicular skeleton that suggest a scratch-digging habit and fossorial adaptations and are envisaged as having used its hypsodont teeth to cut roots and break the substrate, to aid digging with claws (Shockey, Croft & Anaya, 2007).

Kraglievich (1934) and Quintana (1992) suggested that glyptodonts were not functionally suited for digging. However, a geometric morphometric study of the limb bones of five glyptodont species of Miocene and Pleistocene age and comparison with extant armadillos led Vizcaíno et al. (2011) to conclude that were generalized diggers, as modern Dasypodini and Euphractini. Generalized diggers are species that dig short burrows for protection or in search of food and that feed on the surface or just below it by making ‘food probes’ (Abba, Udrizar & Vizcaíno, 2005). In order to asses if glyptodonts were likely producers of the fossil burrows we compared the width of the dorsal carapace and the dorsal carapace height / width ratio with comparable values of the fossil burrows. Dorsal carapace width of Miocene-Pliocene glyptodonts range between 0.40 and 0.77 m (Perea, 2005; Vizcaíno et al., 2011; Zurita et al., 2011), well in the range of horizontal diameter of the fossil burrows. Information on the ratio between carapace height and width for Miocene-Pliocene glyptodonts is incomplete, and similar data for Pleistocene South American glyptodonts (Duarte, 1997; Zurita et al., 2010) average 0.87 (range = 0.78–0.91; n = 4). In our case study, glyptodonts are considered good candidates for constructing the subcircular burrows of the intermediate subpopulation, which are 0.39–0.56 m wide and display an average Dv/Dh ratio of 0.88. Regarding the large dasypodids Macrochorobates and Macroeuphractus, the available body mass estimates suggest a range of 10 to 100 kg (Vizcaíno & Fariña, 1999) and little is known about their paleoecology.

Among the mylodontids, the Scelidotherinae, endemic to South America (McDonald, 1987; Taglioretti et al., 2014); are only represented for the Huayquerian—Chapadmalian SALMAs (late Miocene to early Pliocene) by Proscelidodon, a ground sloth related to open environments with grasslands, under temperate and warm climate (Miño Boilini et al., 2011; Pujos et al., 2012; McDonald & Perea, 2002). A digging habit was inferred for Proscelidodon after a morphofunctional study of a Montehermosian (latest Miocene-early Pliocene) forelimb (Aramayo, 1988). Body mass estimates are only available for Pleistocene scelidotherines (Table 1) and range from 584 to 1,057 kg (De Esteban-Trivigno, Mendoza & De Renzi, 2008; Bargo et al., 2000; Fariña, Vizcaíno & Bargo, 1998). These would be maximum estimates for late Miocene scelidotherines because the primitive Mylodontidae were smaller and there seems to be a trend toward progressively larger sizes in the Pleistocene (e.g., McDonald & Perea, 2002). Large Pliocene-Pleistocene fossil burrows near Mar del Plata city (Buenos Aires province) have been attributed to mylodonts on the basis of the finding of bone remains inside the fill (Frenguelli, 1955) and using the surface ornamentation of the burrows (Zárate et al., 1998; Dondas, Isla & Carballido, 2009).

For the large subpopulation, with a Dh ranging from 1.05 and 1.50 m (9% of cases) and an extrapolated body mass of 700–1,600 kg, the more likely producer is Proscelidodon sp. and, secondarily, the Glyptodontidae.

To summarize, the studied fossil burrows can be attributed to several producers, according to their horizontal diameter. The more likely producers of the studied fossil burrows are: (1) for the small subpopulation, the smaller dasypodids (Doellotatus, Chasicotatus, Proeuphractus and Chorobates) on the basis of body mass, the fossorial habit and architecture of modern dasypodid burrows and, secondarily, Paedotherium minor. (2) For the intermediate and large subpopulations, the Glyptodontidae and Mylodontidae (Proscelidodon sp.) are good candidates as these were the largest representatives of the late Miocene burrowing fauna of the Cerro Azul Formation. The Glyptodontidae were generalized diggers, like modern dasypodids, and exhibited a carapace fitting especially the subcircular burrows. Proscelidodon sp. is also a likely candidate of the elliptical and larger burrows. For the intermediate subpopulation, probably the large dasipodids (Macrochorobates and Macroeuphractus) and Mesotheriinae should be considered.

Taphonomy of burrows

The horizontally laminated and massive fill of the burrows suggest that the material entered the excavation passively, that is after the burrow had been fully excavated, and without any assistance by the digger. The infill also indicates that the burrows were abandoned and received sediments in successive small pulses and large catastrophic events. Although we cannot discard some secondary input of dust by wind, most of the filling of the burrows is related to water transport as indicated by the well laminated and normal graded siltstone to mudstone laminae (Figs. 10A–10C, 10F). Laminated intervals are linked to successive pulses of sediment-laden water that eventually ponded in the terminal tracts of the burrows. This is in agreement with the interpretation by Imbellone, Teruggi & Mormeneo (1990) of similar Quaternary burrows and experimental results by Woodruff & Varricchio (2011). Experiments by Woodruff & Varricchio (2011) indicate that well-laminated fills were obtained by adding small amounts of sediment-water mixtures entering at a low angle (5°) into the burrow. In contrast, en masse pouring of the sediment-water mixture at high angle (30°) produced a massive and poorly sorted sediment fill, whereas en masse pouring at a low angle (5°) produced thicker graded beds (Woodruff & Varricchio, 2011). En masse filling experiments also produced “arcuate structures” (Woodruff & Varricchio, 2011) that are very similar to the pseudomeniscate structures described herein. Both features are comparable to “arcuate surface ridges” produced in experimental debris flows that reflect the pulsatory nature of experimental and natural debris flows (Major, 1997). The experiments by Woodruff & Varricchio (2011) lend further support to the interpretation of the massive intervals as result of catastrophic input of large volume of unsorted sediment. As the burrows are related to an essentially flat landscape and no fluvial channel deposits were observed in any of the studied localities, the sediment pulses should be related with rainfall. We speculate that one or more laminae may result from moderate to heavy precipitation events. In contrast, massive intervals can be related to single heavy downpours producing sheet flooding, which can generate hyperconcentrated flows (e.g., Smith & Lowe, 1991) capable of transporting enough material to fill, at least, the terminal portion of a burrow in a single event. High-energy sheet floods can also saturate burrow walls and produce roof collapse.

Our studies also support the generalization that well laminated burrow fills will not contain remains of the producer and that massive fills of the whole burrow or most of the lower part have a greater chance of containing remains of the tetrapod that dug the burrow, as proposed by Scognamillo (1993) and Groenewald, Welman & MacEachern (2001). For the laminated burrow fills, the most likely scenario is that the burrow was vacated or, if the animal died inside, it may result scavenged and/or weathered, which produces incomplete and disarticulated remains. In the case of a massive fill, both live entombing (Scognamillo, 1993; Groenewald, Welman & MacEachern, 2001) and fast burial after death (Woodruff & Varricchio, 2011) are required to produce a nearly articulated and complete skeleton. Massive fills in the upper half of the burrow will not preserve remains of its producer.

The episodic nature of the filling processes is evidenced by the laminated fill and further supported by the presence of footprints in the surface of some laminae and the cluster of bee cells (?Rosellichnus isp.) found inside the fill. These trace fossils suggest that partially filled burrows with a surface communication were explored or reoccupied by other tetrapods and used by bees to nest. Alternatively, the bee cells may be constructed after the complete filling of the burrow in the soil profile. Among the presumed producers of burrows of intermediate size, the outline and size of the footprints match those of Pleistocene glyptodonts but are quite different from those of ground sloths (compare Aramayo et al., 2015). Disruption of laminae composing the fill of the burrows is explained by drying and cracking of mud after a flood event, whereas synsedimentary faulting can be related to trampling by tetrapods.

Attaichnus kuenzelli occur profusely in the SG locality, in some cases very close, but never were cut by a large mammal burrow. These relationships suggest that the producers of the burrows were apparently not foraging on A. kuenzelli, considered a nest chamber of leaf-cutting ants (Genise et al., 2013).

Paleoecological and paleoenvironmental meaning

Detailed inferences about the paleoecological and paleoenvironmental meaning of the studied large mammal burrows can be gained through sedimentological study of the hosting rocks, analysis of orientation of fossil burrows and considering the associated trace fossils. This information, along with the potential producers will help to understand the reason for acquisition of burrowing habits in large late Miocene mammals.

Sedimentology

Thick, monotonous, massive continental successions of siltstone showing moderate to good sorting with associated paleosols, as those described for the Cerro Azul Formation, are typical of loess deposits, whose dominantly eolian origin is well established (e.g., Johnson, 1989; Pye, 1995). The presence of pedogenic calcite is indicative of well-drained soil profiles in sub-humid, semi-arid, and arid climates with low rainfall (less than 800 mm/yr) and high evapotranspiration (see review in Sheldon & Tabor, 2009). Previous estimation of mean annual precipitation for the development of the paleosols of the Cerro Azul Formation is 449 ±147 mm (Cardonatto et al., 2016). Paleosols showing a Bt horizon and blocky or prismatic peds can be compared with mollisols (Cardonatto et al., 2016). Some paleoenvironmental constraints can also be derived from the composition of the mammal fauna, and the stable isotopic composition of enamel teeth. Vertebrate remains of the Cerro Azul Formation, mainly notoungulates and rodents, suggest that these sediments were deposited in open landscapes like steppes or herbaceous plains (Montalvo et al., 2008). Carbon isotope composition from late Miocene herbivorous enamel teeth from Salinas Grandes de Hidalgo and nearby localities indicates a dominance of C3 plants in lowland areas (MacFadden, Cerling & Prado, 1996), which are favoured in climates with a cool growing season (Ehleringer, Cerling & Helliker, 1997)

Orientation of burrows

Comparison with orientation data from modern Dasypodidae burrows can help to interpret the orientation pattern of fossil burrows. As xenarthrans are imperfect homeotherms, their body temperatures do change with the environment (e.g., McNab, 1980; McNab, 1985). It has been suggested that the burrow entrance orientation of armadillos avoid prevailing winds and both uniform and preferential orientation has been documented (e.g., McDonough & Loughry, 2008). The cases of no preferential orientation are related to the invasive armadillo Dasypus novemcinctus from southern USA (Texas, Alabama, Oklahoma) and Belize (Clark, 1951; Zimmerman, 1990; Platt, Rainwater & Brewer, 2004; Sawyer et al., 2012). All these cases are mostly related to forested areas. Studies documenting a preferred orientation of Dasypodidae burrows are from Argentina, Uruguay and Brazil, involving open environments and several species (Crespo, 1944; Carter & Encarnaçao, 1983; González, Soutullo & Altuna, 2001; Abba, Udrizar & Vizcaíno, 2005; Ceresoli & Fernandez-Duque, 2012). The pioneer study by Crespo (1944) included three localities from western Argentina, ranging from 27°37″S to 34°13″S with annual precipitation ranging from less than 200 mm to 500 mm. The vegetation ranges from low bushes, to shrubland and psammophilous grassland with sparse trees. These localities belong to the Monte and Espinal biogeographic provinces (e.g., Roig, Roig-Juñent & Corbalán, 2009) and the included armadillo species are: Chaetophractus vellerosus, C. villosus and Zaedyus pichiy. A compilation of the entrance orientation data from the three localities of Crespo (1944) suggests a dominant entrance orientation toward the west (Fig. 9D). This distribution is remarkably similar to the fossil burrows if we assume that entrance orientation was at 180°of dipping azimuth (Fig. 9C). Dominant surface wind patterns in northern Argentina are humid and sometimes hot winds from the east and north (e.g., Barros et al., 2015), whereas cold winds are from the south. In consequence, the orientation pattern described by Crespo (1944) from open environments of the semiarid region of Argentina can be interpreted as preferential orientation of entrances avoiding dominant hot and cold winds. Similar patterns of armadillo burrow entrance orientation avoiding prevailing winds were documented by Carter & Encarnaçao (1983) in Minas Gerais, Brazil; González, Soutullo & Altuna (2001) in Uruguay (Fig. 9E); Abba, Udrizar & Vizcaíno (2005) in Buenos Aires province of Argentina (Fig. 9F); and Ceresoli & Fernandez-Duque (2012) in Formosa province, northern Argentina. Alternative explanations for this preferential orientation are that, as the armadillos seek food following an odour in the wind, they tend to approach a site from downwind and dig in the lee side (Carter & Encarnaçao, 1983) and to maximize sun exposure during cold winters (Ceresoli & Fernandez-Duque, 2012). In particular, the most adequate example to evaluate the orientation of the fossil burrows is the data from dasypodid burrows by Crespo (1944), which were collected in open semiarid settings similar to those of the late Miocene of central Argentina. In consequence, it is possible to propose that the late Miocene wind pattern of central Argentina was similar to the present one with hot winds from the east and north and cold winds from the south.

Associated trace fossils

The trace fossil assemblage of the Cerro Azul Formation is of low diversity and abundance and dominated by insect trace fossils (Celliforma, Rosellichnus, Fictovichnus, Rebuffoichnus and Teisseirei), and was compared with the Celliforma ichnofacies (Cardonatto et al., 2016). The Celliforma ichnofacies is typical of well-drained calcareous paleosols developed under low vegetation coverage (Genise et al., 2010; Genise et al., 2016). The reduced size of associated rhizoliths suggests that the vegetation was dominated by scrubs with minor participation of herbaceous plants.

The local occurrence of cemented Coprinisphaera at LLP and additional occurrences of fossil dung-beetle brood balls (Quirogaichnus coniunctus Laza, 2006) from the formation in a nearby locality (Laza, 2006) is indicative of the presence of the Coprinisphaera ichnofacies, suggesting herbaceous communities and wetter climatic conditions (Genise et al., 2016) for the easternmost locations of the formation.

Burrowing habits in large late Miocene mammals

Mammal burrows are typically constructed as shelters from environmental extremes and predators, and also for food storage, foraging and reproduction (e.g., Reichman & Smith, 1990; Kinlaw, 1999). From these common uses of burrows, protection from environmental extremes and predators are more likely for the studied fossil burrows and no evidence supporting the remaining functions is available. Top predators during deposition of the Cerro Azul Formation are the Phorusrhacidae (Cenizo, Tambussi & Montalvo, 2012; Vezzosi, 2012) that occupied the role of large carnivorans, as well as the Sparassodonta (Goin, Montalvo & Visconti, 2000).

However, the main factor controlling the occurrence of large mammal burrows during the late Miocene (Fig. 1) is herein related to environmental changes. It has been suggested that different mammal groups acquired fossorial habits during the Cenozoic as a response to the expansion of open, savanna-like environments under cold, dry and seasonal climates (Nevo, 1979; Nevo, 1995; Nevo, 2011). During the late Miocene (the Huayquerian SALMA), southern South America experienced a global cooling as response to the increase in the Antarctic ice sheet (Zachos et al., 2001) and the uplift of the Andes (e.g., Amidon et al., 2017), which favored cold and seasonally dry climatic conditions. This regional framework is confirmed by the inferences on the sedimentology, faunal remains and invertebrate ichnology of the studied succession. This evidence suggests open environments, with well-drained soils and low vegetation coverage, and a semiarid and seasonal climate. Considering that the more likely candidates for the largest burrows are xenarthrans (Glyptodontidae and Mylodontidae), which are imperfect homeotherms (e.g., McNab, 1980; McNab, 1985), the necessity and convenience for excavating an underground refuge is clear. In addition to escape from predation, these animals used burrows to avoid extremely cold or warm climatic conditions in order to conserve energy and water, and to breed because of the particular physiology of xenarthrans (Vizcaíno et al., 2001).

Conclusions

Subcylindrical structures with a laminated fill occurring in a late Miocene loess-paleosol sequence from central Argentina are interpreted as burrows of fossorial mammals. The burrows occur associated with calcareous paleosols developed under a semiarid climate in a flat landscape. A detailed analysis of more than one hundred structures permitted to conclude that:

1. The more common geometry is a shallow ramp with a rounded end, lacking bifurcations.

2. The ample variation of the horizontal diameter of the burrows, along with cross-sectional shape and inferred body masses suggest that several producers were involved.

3. The smaller burrows (Dh = 0.15–0.34 m, 8% of cases, body mass ranging from 1 to 13 kg) were produced by small dasypodids and, secondarily, by the notoungulate Paedotherium minor.

4. For the dominant burrows exhibiting an intermediate (Dh = 0.39–0.94 m, 83% of measured burrows, producer body mass of 37 to 438 kg), and large horizontal diameter (Dh = 1.05–1.50 m, 9 % of measured burrows, producer body mass of 700–1,600 kg), the Glyptodontidae and Mylodontidae (Proscelidodon sp.) are the best candidates. The Glyptodontidae are related to the subcircular burrows of intermediate size and Proscelidodon sp. would be the producer of the elliptical and largest burrows.

5. About 10% of the burrow fills contained fragmentary, disarticulated, abraded and weathered bone remains that were introduced into the burrows by aqueous currents and do not belong to the excavator of the burrow.

6. After abandonment, the burrows received sediments from the surface that progressively filled the structure. The filling process was produced dominantly by several pulses of sediment laden currents related to moderate rains (well laminated fill) and en masse input of dense sediment-laden currents related to heavy rains producing sheet flooding (massive fill). During filling, the abandoned burrows were visited or reoccupied by other tetrapods and used by bees to nest,

7. In general, it is not considered likely that well laminated burrow fills will contain remains of the producer, whereas massive fills have a greater chance of containing remains of the tetrapod that dug the burrow.

8. The main factor explaining the appearance of large mammal burrows in the late Miocene are environmental changes, including the appearance of open environments with low vegetation and semiarid and seasonal climate.

Supplemental Information

Article S1 Source of information of Fig. 1

Plot of average horizontal diameter of fossil burrows vs age of the hosting rock. Table containing raw data for Fig. 1 and literature source. Image credit: María Cristina Cardonatto and Ricardo Néstor Melchor.

Click here for additional data file.

Article S2 Raw data for geometric morphometric analysis of burrow cross section

Compressed .TPS file.

Click here for additional data file.

Article S3 Systematic paleontology

Systematic assignation of bone remains found inside burrow fills. Credit: María Cristina Cardonatto.

Click here for additional data file.

Figure S1 Bone remains found within burrow fills

(A) Proscelidodon sp. (GHUNLPam 18807-1). (B) fragmented metacarpal of Proscelidodon sp. (GHUNLPam 18807-2). (C) Gliptodontidae indet. (MGH-P126/41). (D) Eosclerocalyptus sp. (MGH-P126/34). (E–F) Glyptodontidae indet. (MGH-P126/37). (G) Doellotatu s sp. (MGH-P126/40). (H) Paedotherium minor (MGH-P126/39). (I) Mesotheriinae indet. (MGH-P126/42). Scale bar: 1 cm. Image credit: María Cristina Cardonatto.

Click here for additional data file.

Table S1 Burrow fills registered in the late Miocene Cerro Azul Formation

References: SC: slightly curved, SI: sinuous, C: “C” shaped, E: elliptical, S: subcircular, PC: planoconvex, * approximate value, () minimum value. Dh, horizontal diameter; Dv, vertical diameter. Body mass estimation after Wu et al. (2015). Wu NC, Alton LA, Clemente CJ, Kearney MR, White CR. 2015. Morphology and burrowing energetics of semi-fossorial skinks (Liopholis spp.). The Journal of Experimental Biology, 218: 2416–2426. Credit: María Cristina Cardonatto.

Click here for additional data file.

We are indebted to Fátima Mendoza Belmontes for help during field work, Claudia Montalvo for guidance with the taxonomy of vertebrate remains and for appropriate comments on an earlier version of the manuscript, Marcelo Zárate for noticing about the fossil burrows of the LLP locality, Ricardo Bonini for field permit, and María F. Vera Candioti for help on morphometric analysis.

Additional Information and Declarations

Competing Interests

Author Contributions

Field Study Permissions

Data Availability

The authors declare there are no competing interests.

María Cristina Cardonatto conceived and designed the experiments, performed the experiments, analyzed the data, contributed materials/analysis tools, prepared figures and/or tables, authored or reviewed drafts of the paper, approved the final draft, and prepared the Systematic Paleontology section.

Ricardo Néstor Melchor conceived and designed the experiments, performed the experiments, analyzed the data, contributed materials/analysis tools, prepared figures and/or tables, approved the final draft, and conducted sedimentologic study.

The following information was supplied relating to field study approvals (i.e., approving body and any reference numbers):

Field work was approved by the Dirección Provincial de Museos y Preservación Patrimonial under the project “Vertebrados del Mioceno tardío-Plioceno en el área de las lagunas encadenadas del oeste de la provincia de Buenos Aires. Aportes a la bioestratigrafía del Cenozoico tardío de la Región Pampeana” directed by Ricardo Bonini.

The following information was supplied regarding data availability:

The raw data are provided in the Supplemental Files.

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
