# Peer review of "Large mammal burrows in late Miocene calcic paleosols from central Argentina: paleoenvironment, taphonomy and producers"

_PeerJ, doi:10.7717/peerj.4787_

## Round 0.1 · original submission · Minor Revisions

Dear Dr Cardonatto

Your manuscript #25129, entitled "Large mammal burrows in late Miocene calcic paleosols from central Argentina: palaeoenvironment, taphonomy and producers " which you submitted to PeerJ has been reviewed by three reviewers and myself.

All reviewers consider that your contribution is suitable for publication and should be accepted after minor revision, a conclusion I agree. In this context, the reviewers have pointed out several changes concerning misspellings, and rewording, among others, and both Reviewer #1 and #3 had included them in their annotated manuscript. Particularly, I strongly suggest you to pay attention to the confusing use of "burrow fill" as a synonym of the burrow itself in several paragraphs of your Ms, as pointed out by Reviewer #2, as this can led to misinterpretations of your data.

Therefore, I am requesting that you address the suggestions mentioned above and resubmit your manuscript to PeerJ.

Looking forward to receiving your revision.

Sincerely, Claudia Marsicano

·

Basic reporting

The manuscript is well written, the English grammar is, in some cases, rather old-fashioned so I have edited it with track change to bring it more into line with modern scientific reporting.

The literature review is exhaustive and up to date.

The article is structured according to current convention for scientific reports. The raw data is in the manuscript as well as supplementary information

The analysis and interpretation of the burrow structures is sound and is a good example of how morphometric landmarks analysis can be successfully applied to field outcrops.

Experimental design

The research topic is within the scope of Peer J as it is based on empirical data collected in the field- analysed and synthesised with a modern statistical approach to prove the origin and preservation mode of these enigmatic tubular structures interpreted as mammalian burrow casts. The authors go in to discuss the palaeoenvironmental significance of these structures which is a definite advance in our knowledge to date.

The methods are fully and succinctly explained

Validity of the findings

The interpretations are based on comparison of the study set with other ancient burrow casts as well as modern fossorial mammals, The presence of bone fossils in the casts is as an indicator of the possible burrower is treated with the proper circumspection.

The two sedimentary facies are involved in infilling of the burrows is novel and useful information to be tested on similar occurrences elsewhere. The sedimentological interpretations are sound and backed up with the relevant references.

Linking the occurrence of burrow casts to rapid environmental change is not new, but there is always scope for such field based studies to back this up.

Additional comments

My detailed comments are in track change

Overall a very competent and informative study with something new to say about the infilling sequences and the environmental significance of vertebrate burrow casts in the fossil record.

I would like to see the outcrop area plotted on the map
I would also like to see a longitudinal profile of a generalised burrow cast showing the thickness relationships of the two facies and the onlapping with the burrow walls both down burrow and across burrow

·

Basic reporting

The English grammar used through the manuscript is clear and professional. However, as I am not a native English speaker my grammar corrections are limited.
The introduction and background are well presented and completed with actualized literature. The figures are relevant, high quality, and well labeled, and the raw data is supplied.

Experimental design

The research is original, with primary data analysis. Although, the structures were partially and briefly described (Genise et al., 2013), but a detailed description and analysis was missing. The research questions are clearly stated, relevant, and meaningful and followed throughout the manuscript. The technical methods used to analyze the data are modern, proper and well described to replicate it.

Validity of the findings

The manuscript is a good example of how to deeply analyze a set burrows of assorted morphologies and sizes. The evaluation of its likely producers is exhaustive and the analysis of the taphonomy of the burrows to finally infer its paleoecological and paleoenvironmental meaning its also very well performed.

Additional comments

I found one minor aspect to be consider through the manuscript. The authors talk continuously about the features of the “burrow fills”, as referring to the main burrow structure. I understand that the burrow fill is one of the features of the burrow itself, even though that is what is mostly observable. The sharp contact between the host rock and the burrow fill is actually the fossil burrow. In some cases could result valid the use of burrow fill as referring to the whole structure (Line 189), but in others it doesn’t (line 19-21). Please consider revising its use in the whole manuscript.
For example, in line 19-21 “Geometric morphometric analysis of transverse cross-sections support the distinction of subcircular and elliptical (horizontally flattened) fills.” What is subcircular or elliptical is the burrow itself and not the burrow fill.
Line 189. “Description of large burrow fills”. Could result correct if you are describing the burrow fill, But, if is used as synonym of burrow and you are describing other attributes of burrows, as the burrow architecture, it results tricky.

·

Basic reporting

No comment.

Experimental design

No comment.

Validity of the findings

No comment.

Additional comments

This is an interesting and professionally designed study that significantly contributes to the evolutionary paleoecology of fossorial behavior, and, of course, is a very detailed and comprehensive study of the Argentinean occurrence. The ms well matches the requirements of PeerJ in all relevant aspects. I do not see potential to improve the paper except for minor formal mistakes or inconsistencies (see marked pdf).
I am just wondering how long it may have taken to completely fill especially the large-sized burrows. Is there any chance to constrain the duration of this process? If there was much time it is remarkable that there seems almost no evidence for reoccupation and restructuring of the primary burrows.
Well done!

Sebastian Voigt, 2018-04-07

---

## Round 0.2 · Minor Revisions

Dear Dr. Cardonatto,

During a final revision of your Ms, I realized that there is some detailed information missing about your analysis. The analysis´s steps should be described in sufficient detail to allow replication from the raw format. Accordingly, I suggest you to include in the Supplementary Information the raw data of your 2D analysis, as the .TPS files. Moreover, there is not detailed explanation about the landmarks (e.g. type?, how did you define them?). It would be very important if you can accurately describe the methodology behind your analysis in the Materials & Methods section.

Finally, I would suggest to include a Table with the data used to construct Figure 1, beyond a reference list.

I am requesting that you revise and include the information mentioned above, simply to make your general discussion and conclusions better supported.

sincerely,

Claudia Marsicano

---

## Author Rebuttal · Round 0.2

**Uruguay 151 - (6300) Santa Rosa - La Pampa**
**Tel.: 02954-425166 - 422026 - Fax.: 432679**
**Email:** fexactas@unlpam.edu.ar
**Página Web:** http://www.exactas.unlpam.edu.ar

**UNIVERSIDAD NACIONAL**
**de LA PAMPA**

April 15th, 2018

Dear Dr Claudia Marsicano
Academic Editor, PeerJ
RE: (#2018:02:25129:0:2:REVIEW)

Dear Dr Marsicano:

Thank you for the competent and pertinent reviews. We have carefully revised the manuscript according to reviewers and editor advice. The following are the answers we have made in response to the reviewer's and editor comments:

Reviewer #1
Basic reporting
The manuscript is well written, the English grammar is, in some cases, rather old-fashioned so I have edited it with track change to bring it more into line with modern scientific reporting. The literature review is exhaustive and up to date. The article is structured according to current convention for scientific reports. The raw data is in the manuscript as well as supplementary information. The analysis and interpretation of the burrow structures is sound and is a good example of how morphometric landmarks analysis can be successfully applied to field outcrops.
Thank you for the constructive comments, we have followed the suggestions in order to make the ms according to modern scientific reporting. We also believe that geometric morphometric analysis is a powerful tool to be applied in field studies.

Experimental design
The research topic is within the scope of Peer J as it is based on empirical data collected in the field- analysed and synthesised with a modern statistical approach to prove the origin and preservation mode of these enigmatic tubular structures interpreted as mammalian burrow casts. The authors go in to discuss the palaeoenvironmental significance of these structures which is a definite advance in our knowledge to date. The methods are fully and succinctly explained.
Thank you for your comments.

Validity of the findings
The interpretations are based on comparison of the study set with other ancient burrow casts as well as modern fossorial mammals, The presence of bone fossils in the casts is as an indicator of the possible burrower is treated with the proper circumspection.
Thank you. We make cautionary comments about the interpretation of bone remains found inside fossil burrows, which are not necessarily from the producer.

[Figure]
[Figure]

The two sedimentary facies are involved in infilling of the burrows is novel and useful information to be tested on similar occurrences elsewhere. The sedimentological interpretations are sound and backed up with the relevant references.
Thank you for your comments.

Linking the occurrence of burrow casts to rapid environmental change is not new, but there is always scope for such field based studies to back this up.
We do not link the preservation of the burrows to rapid environmental changes, instead we interpret it as result of normal processes in a stable environment.

Comments for the Author
My detailed comments are in track change
All the changes suggested in the annotated manuscript were introduced, except for the new redaction of the purpose of the paper proposed by Reviewer # 1. In this case we believe that the new redaction involves aspects not included in the ms (for example the mechanism of excavation) and prefer to adhere to the original redaction, although with minor changes, in order to make them clearer.

Overall a very competent and informative study with something new to say about the infilling sequences and the environmental significance of vertebrate burrow casts in the fossil record.
Thank you for your comments.

I would like to see the outcrop area plotted on the map
We modified Fig. 2 accordingly to include the outcrops of the Cerro Azul Formation.

I would also like to see a longitudinal profile of a generalised burrow cast showing the thickness relationships of the two facies and the onlapping with the burrow walls both down burrow and across burrow
We understand that this schematic diagram may be illustrative but prefer not to include this diagram because it would be very speculative with the available information.

Reviewer # 2
Basic reporting
The English grammar used through the manuscript is clear and professional. However, as I am not a native English speaker my grammar corrections are limited.
The introduction and background are well presented and completed with actualized literature. The figures are relevant, high quality, and well labeled, and the raw data is supplied.
Thank you for your comments.

Experimental design

[Figure]

[Figure]

**UNIVERSIDAD NACIONAL**
**de LA PAMPA**

**Uruguay 151 - (6300) Santa Rosa - La Pampa**
**Tel.: 02954-425166 - 422026 - Fax.: 432679**
**Email:** fexactas@unlpam.edu.ar
**Página Web:** http://www.exactas.unlpam.edu.ar

The research is original, with primary data analysis. Although, the structures were partially and briefly described (Genise et al., 2013), but a detailed description and analysis was missing. The research questions are clearly stated, relevant, and meaningful and followed throughout the manuscript. The technical methods used to analyze the data are modern, proper and well described to replicate it.

Thank you for your comments. The paper by Genise et al. (2013) only provided a general and brief description of a few examples of one of the three localities studied in this paper.

Validity of the findings
The manuscript is a good example of how to deeply analyze a set burrows of assorted morphologies and sizes. The evaluation of its likely producers is exhaustive and the analysis of the taphonomy of the burrows to finally infer its paleoecological and paleoenvironmental meaning its also very well performed.

Thank you for your comments.

Comments for the Author
I found one minor aspect to be consider through the manuscript. The authors talk continuously about the features of the "burrow fills", as referring to the main burrow structure. I understand that the burrow fill is one of the features of the burrow itself, even though that is what is mostly observable. The sharp contact between the host rock and the burrow fill is actually the fossil burrow. In some cases could result valid the use of burrow fill as referring to the whole structure (Line 189), but in others it doesn't (line 19-21). Please consider revising its use in the whole manuscript.
For example, in line 19-21 "Geometric morphometric analysis of transverse cross-sections support the distinction of subcircular and elliptical (horizontally flattened) fills." What is subcircular or elliptical is the burrow itself and not the burrow fill.
Line 189. "Description of large burrow fills". Could result correct if you are describing the burrow fill, But, if is used as synonym of burrow and you are describing other attributes of burrows, as the burrow architecture, it results tricky.

We realized the mistake and revised the entire manuscript to avoid this confuse usage.

Reviewer # 3
Basic reporting
No comment.

Experimental design
No comment.

Validity of the findings
No comment.

Comments for the Author

This is an interesting and professionally designed study that significantly contributes to the evolutionary paleoecology of fossorial behavior, and, of course, is a very detailed and comprehensive study of the Argentinean occurrence. The ms well matches the requirements of PeerJ in all relevant aspects. I do not see potential to improve the paper except for minor formal mistakes or inconsistencies (see marked pdf).
All the changes suggested in the annotated manuscript were introduced

I am just wondering how long it may have taken to completely fill especially the large-sized burrows. Is there any chance to constrain the duration of this process? If there was much time it is remarkable that there seems almost no evidence for reoccupation and restructuring of the primary burrows.
Well done!
Thank you for your comments. About the length of the infilling processes, the only that we can infer is that was roughly contemporary with the sedimentation of the formation. It probably took several years. We do have evidence of re-occupation in the form of footprints and bee cells.

Editor comments
Your manuscript #25129, entitled "Large mammal burrows in late Miocene calcic paleosols from central Argentina: palaeoenvironment, taphonomy and producers " which you submitted to PeerJ has been reviewed by three reviewers and myself.
All reviewers consider that your contribution is suitable for publication and should be accepted after minor revision, a conclusion I agree. In this context, the reviewers have pointed out several changes concerning misspellings, and rewording, among others, and both Reviewer #1 and #3 had included them in their annotated manuscript.
We have introduced all the suggested changes.

Particularly, I strongly suggest you to pay attention to the confusing use of "burrow fill" as a synonym of the burrow itself in several paragraphs of your Ms, as pointed out by Reviewer #2, as this can led to misinterpretations of your data.
We revised the entire manuscript to avoid this confuse usage.

I have no doubt that all the suggested changes have improved ths ms and hope that the new version meet your requirements.
Sincerely yours,

María Cristina Cardonatto

---

## Round 0.3 · accepted · Accept

Dear Dr Cardonatto,

It is a pleasure to accept your Ms # 25129, co-authored with R. Melchor, entitled "Large mammal burrows in late Miocene calcic paleosols from central Argentina: palaeoenvironment, taphonomy and producers".

Thank you for your fine contribution. We look forward to your future contributions to the Journal.

sincerely,

Claudia Marsicano

#